# Autoimmune Complications in Hematologic Neoplasms

**DOI:** 10.3390/cancers13071532

**Published:** 2021-03-26

**Authors:** Wilma Barcellini, Juri Alessandro Giannotta, Bruno Fattizzo

**Affiliations:** 1Hematology Unit, Fondazione IRCCS Ca’ Granda Ospedale Maggiore Policlinico, 20122 Milan, Italy; juri.giannotta@policlinico.mi.it (J.A.G.); bruno.fattizzo@unimi.it (B.F.); 2Department of Oncology and Oncohematology, University of Milan, 20122 Milan, Italy

**Keywords:** autoimmune hemolytic anemia, immune thrombocytopenia, chronic lymphocytic leukemia, lymphoma, myelodysplastic syndrome, chronic myelomonocytic leukemia, myeloproliferative neoplasms, systemic lupus erythematosus, rheumatoid arthritis, vasculitis

## Abstract

**Simple Summary:**

This review analyses the occurrence and clinical characteristics of autoimmune cytopenias and other autoimmune diseases in various lymphoid and myeloid neoplasms. Autoimmune hemolytic anemia and immune thrombocytopenia are observed in about 10% of chronic lymphocytic leukemia and with higher frequencies in certain subtypes of non-Hodgkin lymphoma. At variance, they occur in less than 1% of myelodysplastic syndromes and chronic myelomonocytic leukemia. Autoimmune diseases are described in up to 30% of myeloid and lymphoid patients, and comprise several heterogeneous conditions, such as systemic lupus erythematosus, rheumatoid arthritis, vasculitis, thyroiditis, acquired hemophilia, thrombotic thrombocytopenic purpura, and anti-phospholipid syndrome. Both autoimmune cytopenias and other autoimmune diseases are observed in about 10% of patients receiving hematopoietic stem cell transplant or treatment with new checkpoint inhibitors. All these autoimmune complications may be difficult to diagnose and manage in patients with hematologic cancers, and may negatively impact on outcome.

**Abstract:**

Autoimmune cytopenias (AICy) and autoimmune diseases (AID) can complicate both lymphoid and myeloid neoplasms, and often represent a diagnostic and therapeutic challenge. While autoimmune hemolytic anemia (AIHA) and immune thrombocytopenia (ITP) are well known, other rarer AICy (autoimmune neutropenia, aplastic anemia, and pure red cell aplasia) and AID (systemic lupus erythematosus, rheumatoid arthritis, vasculitis, thyroiditis, and others) are poorly recognized. This review analyses the available literature of the last 30 years regarding the occurrence of AICy/AID in different onco-hematologic conditions. The latter include chronic lymphocytic leukemia (CLL), lymphomas, multiple myeloma, myelodysplastic syndromes (MDS), chronic myelomonocytic leukemia (CMML), myeloproliferative neoplasms, and acute leukemias. On the whole, AICy are observed in up to 10% of CLL and with higher frequencies in certain subtypes of non-Hodgkin lymphoma, whilst they occur in less than 1% of low-risk MDS and CMML. AID are described in up to 30% of myeloid and lymphoid patients, including immune-mediated hemostatic disorders (acquired hemophilia, thrombotic thrombocytopenic purpura, and anti-phospholipid syndrome) that may be severe and fatal. Additionally, AICy/AID are found in about 10% of patients receiving hematopoietic stem cell transplant or treatment with new checkpoint inhibitors. Besides the diagnostic difficulties, these AICy/AID may complicate the clinical management of already immunocompromised patients.

## 1. Introduction

There is increasing awareness of autoimmune complications in hematologic malignancies. Peripheral autoimmune cytopenias (AICy) such as autoimmune hemolytic anemia (AIHA) and immune thrombocytopenia (ITP) are well known complications of lymphoproliferative disorders (LPD) [1,2]. Other less investigated autoimmune diseases (AID) include various organ-specific disorders targeting endocrine glands, liver, and gut, as well as systemic AID (i.e., systemic lupus erythematosus, SLE, rheumatoid arthritis, RA, and antiphospholipid syndrome, APS). In addition, there is evidence of several autoimmune phenomena, i.e., autoantibodies against different autologous proteins, which may, or may not, anticipate overt disease. Diagnosis of AICy may be challenging in hematologic malignancies, due to overlapping conditions like chemotherapy, bone marrow infiltration, and transfusion support. It is known that the direct antiglobulin test (DAT) or Coombs test for the diagnosis of AIHA may be negative in up to 10% of cases, and that the detection of anti-platelet and anti-neutrophil antibodies has low sensitivity [1,2]. Moreover, AIDs are multi-systemic conditions, frequently referred to different specialists, whose diagnosis mainly relies on a constellation of overlapping clinical/laboratory features. Importantly, both AICy and AID may have a severe and life-threatening presentation as well as a chronic/relapsing clinical course impacting on patient’s outcome.

Autoimmunity results from a complex interplay of genetic and environmental factors, including infections and drugs. It is known that primary immunodeficiencies have a high frequency of autoimmune disorders along with an aberrant immune response against pathogens [2,3]. Furthermore, autoimmune patients have a higher susceptibility to infections, possibly as a result of immunosuppressive therapy, triggering a vicious circle that further worsens immune dysregulation. Drugs are a well-recognized cause of AID, including the historical reports of procainamide and hydralazine for SLE, and methyldopa for AIHA. More recently, the new biological modulators, such as inhibitors of tumor necrosis factor (TNF)-α, other cytokine inhibitors, and immune checkpoint inhibitors have been associated with several autoimmune complications. Solid organ and hematopoietic stem cell transplants (HSCT) have an increased risk of AID/AICy, deriving both from the deep immunological storm induced by the transplant itself and from the therapy-induced immunodepression and consequent infections [4,5]. In this review we will describe the occurrence of AID/AICy in lymphoid and myeloid neoplasms, focusing both on the well-known peripheral cytopenias and on the more ignored extra-hematological diseases. Moreover, the impact of HSCT and new cancer therapies will be reviewed.

## 2. Mechanisms of Autoimmunity

Autoimmunity results from the breakdown of both central and peripheral tolerance against self-antigens. The former occurs as negative selection by eliminating autoreactive immune effectors during the maturation of the immune system in the bone marrow and the thymus and include B-cell and T-cell tolerance. Following interaction with self antigens, B-cells undergo clonal deletion via apoptosis and/or induction of anergy. Similarly, T-cells may become tolerogenic after the interaction with self antigens and MHC class I and class II molecules. Moreover, they may undergo a conversion to regulatory T cells (Tregs) that are able to suppress self-reactive T cells. Tregs act through down-regulation of interleukin (IL)-2 and secretion of the inhibitory/tolerogenic cytokines IL-10 and transforming growth factor (TGF)-β. Peripheral tolerance includes the complex immunologic pathways that may inactivate residual autoreactive T-cells after their egression from primary lymphoid organs (thymus and bone marrow), and occurs again by clonal deletion, anergy induction, or activation of Tregs. Other key regulators of autoimmunity are antigen presenting cells (APCs), which contribute to Treg conversion and interaction with the T cell receptor [6]. Autoimmunity recognizes a genetic background and several environmental factors (Figure 1). Among the former, HLA molecules and polymorphisms of genes encoding for various cytokines have been closely associated with different autoimmune diseases. More recently, several regulatory functions have been attributed to microRNA (miRNAs) that repress the expression of target genes via post-transcriptional mechanisms. A dysregulation of several miRNAs has been reported in autoimmune diseases, with consequent pathogenic and potential therapeutic implications [7]. Moreover, virus-derived miRNAs have also been implied in the pathogenesis of autoimmunity, through their interplay with the host genome and consequent genomic instability, as reported for the Epstein-Barr virus (EBV) [8]. Several viruses, bacteria and parasites can induce autoimmunity via different mechanisms. The most known is molecular mimicry, in which B and T cells, activated as a result of an infection, recognize self-molecules that are similar to those from infectious agents. Other mechanisms include non-specific polyclonal activation, and release of intracellular antigens after cell death (known as cryptic antigens). Additional evidence for the role of infectious agents derives from recent reports on an altered microbiota in autoimmune and rheumatic diseases [9]. A further theory of autoimmunity involves the “idiotype-anti-idiotype network”, wherein there is a breakdown of the naturally-existing antibodies capable of neutralizing self-reactive antibodies. Finally, the “forbidden clone” hypothesis proposed more than 60 years ago by Burnet has received a renewed attention, particularly for hematologic neoplasms. In fact, clonality of B cells and plasma cells has been described in a number of autoimmune disorders supporting the notion of autoreactivity within the B cell lineage [10]. The role of B lymphocytes is further sustained by the existence of a subset of B regulatory lymphocytes that produces IL-10 and TGF-β, thus adding an immunosuppressive function to these cells usually considered mere effectors as antibody-producing cells.

## 3. Autoimmune Complications in Lymphoid Neoplasms

The above mentioned pathogenetic mechanisms of autoimmunity may be magnified in hematologic malignancies, particularly LPD, where the T and B-cell compartments are not only “neoplastic” but also highly dysfunctional. As a matter of fact, both AICy and organ/non-organ specific AID, may complicate LPD course from diagnosis to disease progression [11] in up to 50% of cases, although unspecific immune activation has to be distinguished from overt autoimmunity. In addition, LPD therapy, including HSCT and the use of novel drugs (i.e., checkpoint inhibitors, CPIs, chimeric antigen receptor T cells, CART, idelalisib, etc.) may further trigger autoimmunity, and will be discussed in a dedicated paragraph.

### 3.1. Autoimmune Cytopenias (AICy) Complicating Lymphoproliferative Diseases

As shown in Table 1, chronic lymphocytic leukemia (CLL) and non-Hodgkin lymphomas (NHL) are the most frequent LPD associated to AICy, and therefore the most studied, whilst Hodgkin lymphoma (HL), Castleman disease (CD), and large granular lymphocyte disorders (LGL) are less frequently reported, in keeping with their lower prevalence.

#### 3.1.1. AICy in CLL

CLL leukemic B-cells show impaired apoptosis, are unable to efficiently produce immunoglobulins, may function as antigen presenting cells, and release a variety of inflammatory cytokines contributing to autoimmune phenomena [1]. In CLL, AIHA is the most frequent form (7–10% of cases), followed by ITP (1–5%), and rarer entities such as pure red cell aplasia (PRCA, <1%) and autoimmune neutropenia (AIN, 0.17%).

It is worth mentioning that the prognosis of AICy in this setting is more favorable than that of cases of cytopenia due to a massive CLL bone marrow infiltration. Regarding AIHA, the presence of unmutated IGHV status, stereotyped IGHV frames, and unfavorable cytogenetics with chromosome 17p and/or 11q deletions represented a clear risk factor, as did the down-regulation of certain miRNAs (miR-19a, miR-20a, miR-29c, miR-146b-5p, miR-186, miR-223, miR-324–3p, miR-484 and miR-660), known to be involved in autoimmune phenomena [1,11].

#### 3.1.2. AICy in NHL

In NHL AIHA prevalence is of about 2–3%, but increases in certain histotypes, particularly angioimmunoblastic T-cell lymphoma (up to 19%) [5]. Additionally, 50% of patients with marginal zone lymphoma developed autoimmunity, mainly AIHA or ITP [12]. It should be noted that cold agglutinin disease (CAD), sustained by monoclonal cold reactive IgM autoantibodies, is almost invariably associated with a CD5+CD20+ B cell clone, often hardly distinguishable from other NHL [2]. Finally, in NHL some case reports of PRCA and aplastic anemia (AA) have also been described, frequently associated with previous chemotherapy, HSCT, and infections as possible triggers [5].

#### 3.1.3. AICy in HL

AIHA and ITP are very rare complications in HL (<1% of patients) [13], mainly occurring in advanced stages, but possibly preceding HL diagnosis in some cases [14]. Recently, some Authors observed that patients with HL have an increased risk of developing AA, calculated as 20 folds higher than that of the general population [25]. The majority developed AA after or concurrently with HL diagnosis (up to 14 years after) and none had HL marrow involvement. Importantly, AA-HL had a very dismal outcome (median survival after AA of 14 months) [25]. Finally, some cases of AIN have been observed in HL, mostly asymptomatic and responding to intravenous immunoglobulins (IVIG) [29].

#### 3.1.4. AICy in Other LPD

As regards rarer LPD, large granular lymphocyte (LGL) disorders often present with cytopenia, either infiltrative or immune-mediated (directed at bone marrow precursors or due to humoral autoimmunity). Distinguishing the various causes is often difficult, as they may coexist. Moreover, polyclonal LGL cells may be observed in bone marrow and peripheral blood of patients with primary AICy without configuring a true LPD [30]. Few reports exist of overt AIHA, mainly refractory to standard treatment, and possibly concomitant to ITP and AIN (Evans syndrome, ES) [15]. Central autoimmunity seems more frequent as demonstrated by various cases of hypomegakaryocytic thrombocytopenia and PRCA responding to IVIG or LGL treatment [20,26]. The rare multicentric Castleman disease (CD), due to human herpesvirus-8 or idiopathic hypercytokinemia (including IL-6), may be rarely complicated by AIHA, ITP, or ES. These cytopenias, are often refractory to first-line steroids and may revert after rituximab and anti-IL6 therapy with tocilizumab [16,21]. In acute lymphoblastic leukemia (ALL), particularly T-ALL, several case reports described the association of AIHA, ITP, and PRCA, with variable triggers (ALL therapy, HSCT and infectious complications) and outcomes. Some Authors warned that, despite rarity, clinical suspicion should be high, since AICy may confound and delay the initial ALL diagnosis [17,22,27]. Finally, some reports described the onset of AICy (including AIHA, ITP, AIN and ES) before, concomitantly or after multiple myeloma (MM), mainly resolving after MM treatment [18,23,24,28].

### 3.2. Other Autoimmune Diseases (AID) in Lymphoproliferative Neoplasms

As shown in Table 2, a plethora of organ and non-organ specific AID may be associated with LPD, mainly RA, SLE, vasculitis, thyroid autoimmune diseases, Sjögren syndrome, and immune-mediated hemostatic disorders.

#### 3.2.1. AID in LGL

The most frequent association is reported for T-LGL disorders, that may be complicated by AID in more than 50% of cases, mainly RA, SLE, Hashimoto thyroiditis, and Sjögren syndrome (SS) [30]. A notable association is Felty syndrome, characterized by the triad of RA, neutropenia, and splenomegaly. In this setting, immunosuppressive therapy may control both diseases, although infectious risk with consequent mortality is very high [39]. In SLE, an LGL infiltrate correlated with presence of cytopenias and with more frequent SLE exacerbations [35]. More recently, Friedman et al. [40] showed that more than 25% of patients with T-LGL may also present SS and suggested that LGL should be sought and excluded in the initial evaluation of SS patients, particularly if cytopenia is present. Contrarily, no significant association with AID has been reported in aggressive NK-cell leukemia [39].

#### 3.2.2. AID in CLL

About 2% of CLL cases had an AID other than AIHA, ITP or ES in a large retrospective study of 964 patients, including SLE, RA, Hashimoto thyroiditis, vasculitis and SS [31]. Some case reports of CLL-associated acquired hemophilia (AH), acquired von Willebrand syndrome (aVWS), and APS have also been published. The latter are of particular clinical relevance to the hematologist since they may be difficult to diagnose in cytopenic patients and after chemotherapy. Prompt substitutive/anti-coagulant therapy is usually required, and a delay may be fatal. Generally, AH and VWS may respond to CLL therapy including rituximab and venetoclax, whilst the use of ibrutinib is discouraged due to bleeding risk [31,42,43].

#### 3.2.3. AID in NHL

The most frequent associations are arthritis with T cell lymphomas, and thyroid autoimmune diseases with diffuse large B cell lymphoma (DLBCL) [32]. In detail, a large Swedish study [33] evaluated 612 DLBCL patients treated from 2000 to 2013 with rituximab plus chemotherapy and found that 106 (17.3%) developed at least one AID, mainly Hashimoto thyroiditis (31.1%) and RA (22.6%). It is still not clear whether the presence of AID may impact on LPD outcome or survival. In older studies before rituximab, AID were associated with an increased rate of all-cause death. Contrarily, in the recent Swedish report, this association was not confirmed, suggesting a protecting role of rituximab which is also effective on AID [33]. Finally, case reports are available also for AH and aVWS (mainly in marginal zone lymphoma and mucosal associated lymphoid tissue, MALT) [42,44].

#### 3.2.4. AID in HL

HL is seldomly associated to AID, particularly autoimmune thyroid diseases (both Hashimoto and Graves’ disease), with a prevalence of 8.6% in a large retrospective study of 519 patients [13]. Other AID included glomerulonephritis and insulin-dependent diabetes mellitus, and more rarely RA, SLE, mixed connective tissue disease, scleroderma, and vasculitis. In HL, AID mainly developed after lymphoma treatment, contrarily to NHL where they generally preceded LPD diagnosis in 70% of cases.

#### 3.2.5. AID in Other LPD

Concerning rarer associations, CD and ALL may be complicated by AID with a frequency <1%, with possible various combinations (i.e., RA with pemphigus and ITP, etc.). These cases may confound the initial LPD diagnosis, may be life-threatening (as for cases of catastrophic APS), and may respond to CD and ALL therapies [34,36,38,45,46,47]. Finally, several cases of SLE, RA, vasculitis, AH and aVWS complicating MM and amyloidosis have been reported [37,41]. As in LPD, the pathogenic association of MM plasma cells and autoantibody production paved the way to the use of anti-MM agents, including proteasome inhibitors and anti-CD38 antibodies [48,49].

## 4. Autoimmune Complications in Myeloid Neoplasms

The occurrence of AICy and AID in myeloid neoplasms is less known compared with LPD. AICy are quite rare, being AIHA reported in about 1% of MDS and ITP in few cases of CMML (Table 3). At variance, several AID, mainly vasculitis, RA, SLE, immune-mediated hemostatic disorders, and other organ specific disorders are frequently observed in MDS and chronic myelomonocytic leukemia (CMML) (Table 4).

### 4.1. AICy in MDS

As regards MDS, AIHA mainly occur in the low-risk setting [37], developing one to many years after MDS diagnosis [51]. ITP has been reported in few, isolated cases, often in association with other autoimmune manifestations [50,94]. Also PRCA has been mainly described in the context of low-risk MDS, showing a favorable response to immunosuppressive treatment [63]. PRCA has also been associated with high-risk MDS, where the defect of erythroid precursors can be rather attributed to the higher percentage of blasts [64]. Interestingly, isochromosome 17q and 5q deletion have been recurrently described in MDS cases with PRCA [65].

### 4.2. AICy in CMML and Other Myeloid Neoplasms

ITP is the most frequently reported AICy in CMML, either concomitant or preceding its diagnosis [57,58]. The myeloid disease is generally characterized by low-risk features, mainly CMML-1 with normal karyotypes. ITP-CMML patients have high relapse rates and respond to all treatments commonly used for idiopathic ITP, except for IVIG. Reports of AIHA and PRCA complicating CMML are much rarer [52,53,66], with some concern about possible evolution to acute leukemia after immunosuppressive treatments. Myeloproliferative neoplasms (MPN) may be rarely complicated by AICy, with isolated case reports of AIHA and ITP in Philadelphia (Ph)-negative MPN cases [55,60]. A case of steroid-responsive AIHA has been reported in post-MDS acute myeloid leukemia (AML) [56]. Finally, AIN and steroid-refractory ITP has been documented in patients with acute promyelocytic leukemia [61,62].

### 4.3. AID in MDS

AID are described in up to 20–30% of MDS patients [50,68]. Two thirds of AIDs develop months/years after MDS diagnosis, and mainly involve vasculitis [94]. MDS-vasculitis patients are generally younger with a prevalence of male sex, as compared to MDS without vasculitis [68]. No correlations with specific MDS subtypes have been reported [69], except for Behçet’s-like syndrome and trisomy 8 [70]. Regarding vasculitis subtypes, the most common are polyarteritis nodosa and giant-cell arteritis. The latter displays a milder clinical course compared to the idiopathic form (less frequent headaches, jaw claudication and optic neuropathy). Moreover, MDS-associated Behçet’s-like syndrome has a delayed onset, greater gastrointestinal and oral involvement and reduced ocular manifestations than the idiopathic form [70,71]. Other frequent associations are relapsing polychondritis and SLE, whilst SS, RA, polymyalgia rheumatica, uveitis, pulmonary proteinosis, thyroiditis, and myositis are sporadically reported [50,68,94]. Neutrophilic dermatosis, both isolated and in the context of relapsing polychondritis, may be associated with the 5q-syndrome, worsening MDS prognosis [72]. The impact of AID on MDS prognosis is controversial. Some large retrospective studies suggest a reduced overall survival for MDS patients with AID, especially for cryoglobulinemic vasculitis, due to higher infectious rate [50]. Conversely, other studies reported no impact on survival, even if increased cardiovascular comorbidity has been reported in MDS with AID [69,70,73]. A large multicenter study involving more than 1,400 MDS patients with autoimmunity suggests that AID-MDS may be associated with rather better overall and leukemia-free survival [74,75]. As regards treatment, MDS-associated AIDs appear more steroid-dependent and may benefit from biologics and MDS-specific therapies, i.e., hypomethylating agents [68,69,71]. Finally, isolated case reports of acquired HA and thrombotic thrombocytopenic purpura (TTP) have been reported in MDS patients, often characterized by severe/fatal outcome [81,82].

### 4.4. AID in CMML and Other Myeloid Neoplasms

Systemic AIDs are described in up to 25% of CMML patients, again mainly systemic vasculitis, followed by connective tissue diseases [57,68,76]. The majority of cases are CMML-1, with a progression to acute leukemia comparable with that of patients without AID [57]. Steroids are usually effective, but second-line therapies are needed in about 40–50% of cases [57,68]. Hypometylating agents used for CMML have been successfully used in about two thirds of AID manifestations [57]. Finally, few reports of acquired HA, TTP and APS exist, sometimes with fatal outcome [83,84,85].

As regards MPN, a recent prospective study found that about 8% of 435 patients presented an overt AID at diagnosis of myeloid neoplasm [77]. Those with a concomitant AID were younger, showed lower hemoglobin levels, more frequent splenomegaly, and lower progression-free survival than patients without AID. In chronic myeloid leukemia (CML), a population-based nationwide Swedish study showed an increased prevalence of pre-existing AID compared with non-CML controls [78]. Other small series/case reports of AID associations with MPN include multiple sclerosis, inflammatory bowel disease, and primary biliary cirrhosis pre-existing to hematologic diagnosis [79,80]. In primary MF, various cases of RA, dermatomyositis, and polyarteritis nodosa have been reported [79]. Regarding hematologic AID, few case reports of AH onset in Ph-negative MPN have been described [88]. Reports of overt AID in AML are even scarcer, with fatal cases of catastrophic APS, severe/fatal TTP, AH, and acquired factor VII deficiency [90,91,92,93].

## 5. Autoimmune Complications Associated with HSCT

HSCT is a unique model of “immune system” transplantation from a donor to a host, aiming to induce a “graft versus leukemia/lymphoma” effect, but that may in turn attack the recipient in the so called “graft versus host disease” (GVHD). The “immunologic storm” that may arise can lead to either graft failure and/or devastating autoimmune reactions [4]. Great efforts have been made to prevent GVHD and several conditioning chemotherapies and immunosuppressive regimens have been studied, prior and after the various HSCT types (i.e., HLA identical related or unrelated donor, haploidentical, cord blood transplant). However, GVHD remains one of the main causes of transplant-related mortality. Its clinical features are extremely pleomorphic and may encompass virtually all organs and tissues, so that it is difficult to distinguish GVHD from other autoimmune complications post-HSCT. The latter include AICy, SLE and lupus-like reactions, systemic sclerosis, inflammatory bowel diseases, and autoimmune attack to central and peripheral nervous system [54,95,96]. The emergence of AID after HSCT has been systematically analysed only in patients transplanted for autoimmune diseases reporting a cumulative incidence of 9.8% at five years [97,98]. Most data arise from single case reports and small case series and mainly include AICy. AIHA, ITP, AIN and ES may complicate HSCT with a cumulative incidence varying from 2.6% to 56% depending on HSCT sources (5 to 56% for cord blood SCT versus 2.6 to 7.8% for other HSCT sources), and adult versus pediatric setting. Autoantibodies are produced by the donor immune system against antigens on erythrocytes and platelets produced by the graft itself, and the clinical picture is generally severe [99]. Concerning AIHA, both warm and cold forms are described, the former developing between 6 and 18 months, versus 2 to 8 months for the latter. Risk factors include use of unrelated donor and HLA-mismatch, occurrence of GVHD, use of cord blood, age <15 years, CMV reactivation, alemtuzumab use, and non-malignant condition pre-HSCT. Mortality may be quite high and increases with infections [99]. Post-HSCT AIHA is generally managed with steroids with low efficacy (about 20%) so most authors suggest the addition of rituximab frontline to increase response rates (89 versus 52% responses used in second line). Splenectomy is effective but usually limited to selected cases given the high surgical, infectious, and thrombotic risk. Regarding novel targeted therapies, alemtuzumab, bortezomib, sirolimus, eculizumab, daratumumab, and abatacept have all been used in multi-refractory patients with heterogeneous outcomes [99]. Finally, the passenger lymphocyte syndrome (PLS) may occur if graft lymphocytes produce antibodies against residual host erythrocytes. This syndrome may be severe, is highly prevented by graft manufacturing, and tends to self-limit. Risk factors for PLS are the use of cyclosporine alone for GVHD prophylaxis, use of peripheral blood rather than bone marrow as source of the graft, use of reduced-intensity conditioning, use of a non-genotypically HLA-matched donor, and use of a female donor. Careful transfusion procedures are warranted in transplanted patients, particularly in mismatched cases [99,100]. ITP case reports mainly describe good outcome with steroid treatment, although fatalities may occur, particularly in ES cases. More rarely, AIN has been described in case reports [95].

## 6. Autoimmune Complications Associated with Old and New Anti-Cancer Drugs

Several drugs have been associated with the development of autoimmune complications (AIHA, ITP, AA, SLE, vasculitis, etc.), either due to drug-dependent antibodies that activate an immune response only while the drug is present, and drug-independent antibodies, which drive an autoimmune response in the absence of the offending drug. Concerning anti-cancer therapies, CLL treatment deserves special consideration: single-agent purine analogs (i.e., fludarabine) may induce CLL-AIHA [1] possibly worsening the imbalance between Th17 and T-regs. FC and FCR combinations (fludarabine, cyclophosphamide and rituximab), as well as bendamustine rituximab, seem safer [101,102]. The anti-CD52 monoclonal antibody alemtuzumab led to treatment-emergent ITP in 9% of CLL cases [102], again possibly due to T-cell dysregulation. No specific trials have addressed the role of small molecules (i.e., ibrutinib, idelalisib, and venetoclax) in CLL-AICy. However, ibrutinib treatment was rarely accompanied by AICy onset in registration trials and ad hoc analyses, whilst most patients with AICy prior to ibrutinib also achieved remission of the autoimmune condition [4]. Data on idelalisib are very limited, but a correlation with increased incidence of AID (hepatitis, colitis and pneumonitis) has been reported [5]; finally, only a case report exist about AIHA after treatment with venetoclax [103], whilst no warnings emerged from large clinical trials. Regarding MM treatment, ITP has been described after lenalidomide treatment, often persisting after drug discontinuation but successfully managed with corticosteroids or IVIG [23].

In CML, AICy onset is mainly related to interferon therapy, mostly used in the past [59,104]. Moreover, tyrosine kinase inhibitors (TKI), particularly imatinib, have been related to the development of severe AA, possibly via an immunologic mechanism different from the classical bone marrow toxicity [67]. Imatinib and dasatinib have been also related to the onset of severe immune-mediated thrombotic microangiopathies, namely TTP and atypical hemolytic uremic syndrome, which resolved after drug cessation [86,87]. Regarding Ph-negative MPN, interferon therapy has been associated with TTP in a patient with polycythemia vera [89].

Concerning novel anti-cancer drugs, checkpoint inhibitors (CPIs) reactivate T lymphocytes to recognize cancer cells by blocking cytotoxic T-lymphocyte antigen-4 (CTLA-4) or programmed death-1 (PD-1) but may induce several immune-related adverse effects. These include AICy [105], endocrinopathies, autoimmune attack against nervous system, myositis/dermatomyositis, pneumonitis, colitis, and nephritis. AICy/AID may range from mild to life-threatening and may require the cessation of CPI treatment and the establishment of immunosuppression, usually steroids. The rechallenge of CPIs after the improvement/stabilization of the adverse event remains inconclusive. On the whole, a higher incidence has been reported with CTLA-4 inhibitors (i.e., ipilimumab) versus PD-1 inhibitors (i.e., pembrolizumab and nivolumab) and their association increased the risk. At variance, hematologic adverse events seem more frequent after PD-1 inhibitors and are mainly unilineage cytopenia, or bilineage cytopenia, whilst AH, TTP, eosinophilia, LGL, and hemophagocytic lymphohistiocytosis are rare [106,107]. A meta-analysis of 9324 patients indicated that the incidence of anemia, thrombocytopenia, and neutropenia was 9.8%, 2.8%, and 0.94%, respectively [106], with AIHA possibly having a fulminant course. A recent revision of the database of the Food and Drug Administration revealed a total of 68 AIHA cases: 43 developed after nivolumab, 13 with pembrolizumab, seven with ipilimumab, five with atezolizumab, and 16% of cases had received two CPIs. The median time to AIHA onset was 50 days, four patients had concurrent thrombocytopenia, other four endocrine abnormalities (thyroiditis, adrenal insufficiency or hypophysitis), and three gastrointestinal adverse events (colitis or hepatitis). All episodes were severe, and mortality was as high as 17%, mainly due to multi-organ failure and delayed diagnosis [19]. This may occur due to the high proportion of DAT-negative cases, and to the high number of steroid-refractory cases. Autoimmune endocrinopathies are well described and include thyroid diseases, hypopituitarism, and type-1-like diabetes. The latter, contrarily to other autoimmune complications, do not resolve after CPI interruption, may be life-threatening and often require lifelong hormonal replacement [108]. Neurologic involvement is rare and included central and peripheral nervous system diseases associated with neural specific autoantibodies, vasculitis, granulomatous and demyelinating disorders, with an unfavorable outcome in up to 30% of patients [109]. Renal immune-related adverse events are even rarer, although possibly underestimated due to the broad differential diagnosis of acute kidney injury in patients with cancer (infection, dehydration, urinary tract obstruction, and nephrotoxin exposure) [110]. Finally, it is worth mentioning the occurrence of life-threatening cytopenias after chimeric antigen T-cell infusions in ALL/NHL. These cytopenias are observed in up to 30% of patients, even in clinical remission of the hematologic disease, and may be treatment-refractory [111].

## 7. Autoimmune Phenomena in Lymphoid and Myeloid Neoplasms

Besides AICy and AID, lymphoid and myeloid malignancies can be associated with several, isolated autoimmune phenomena that do not reach clinical significance. In CLL a positive DAT without hemolysis is described in about 15% of patients, with only a fraction developing overt AIHA [112]. Moreover, a more sensitive test (mitogen-stimulated DAT, MS-DAT) was able to disclose a latent autoimmunity in about 50% of CLL cases without a definite prognostic significance [113]. In MDS and CMML, a retrospective case series reported a DAT-positivity in up to 35% of patients [114]. Additionally, more than 50% of low-risk MDS patients with hemolytic features displayed anti-erythroblast autoantibodies in bone marrow cultures using MS-DAT, possibly indicating an autoimmune attack against erythroid precursors [115]. Regarding other autoantibodies, more than half of MDS patients display anti-nuclear antibodies (ANA), anti-phospholipid antibodies (APLA), anti-neutrophil cytoplasmic antibodies, rheumatoid factor and organ-specific autoantibodies [116]. Similarly, ANA, immune complexes, lupus anticoagulant, anti-platelet and anti-erythrocyte antibodies by MS-DAT have been reported in more than 50% of patients with primary MF [79,117]. Interestingly, the presence of APLA correlated with thrombosis in atypical sites in patients carrying the *JAK2V617F* mutation [118]. APLA can be also detected in up to 65% of AML patients, correlating with AML persistence/relapse but not with an increased thrombotic risk [119]. Finally, an increased incidence of autoantibodies (especially ANA) occurring post-HSCT has been reported in the literature, particularly anti-neutrophils (44% of patients), although only few subjects developed neutropenia [120]. Notably, autoimmune phenomena without overt clinical disease can be found also in a small fraction of healthy population (e.g., ANA and rheumatoid factor up to 4% and DAT positivity in 0.1–1%).

## 8. Conclusions

Autoimmune complications are frequent in both lymphoid and myeloid neoplasms, not always easy to diagnose, and frequently severe and life-threatening. AICy are particularly common in LPD, with frequencies up to 10% in CLL or even higher in certain subtypes of NHL. At variance, AICy in myeloid neoplasms are much rarer (less than 1%), mostly observed in low-risk MDS and CMML (especially ITP). Whilst AIHA and ITP are known complications, the rarer AIN, PRCA, and AA should be considered in the diagnostic algorithm of the cytopenic patient, taking into account bone marrow infiltration and/or peripheral destruction. Pitfalls for the latter include DAT-negative AIHA and the low sensitivity of anti-platelet and anti-neutrophil autoantibodies, whereas the kinetics of clinical presentation and response to steroid/IVIG may help the diagnosis. CLL/lymphoma-specific therapy is generally required in relapsed/refractory AICy cases or in patients with progressive LPD-disease.

Regarding AID, their identification, diagnosis and treatment in the context of lymphoid and myeloid disorders is even more difficult. This is due to their great clinical heterogeneity and variable organ involvement, the lack of specific diagnostic tests, and the unfamiliarity of hematologists with these protean diseases. Moreover, the clinical picture of AID can be confounded by signs/symptoms of the hematologic disease and/or its specific treatments. Notwithstanding these difficulties, systemic (i.e., SLE, RA, SS) and organ-specific AID are reported in a considerable proportion of LPD, particularly LGL leukemia. The frequency of AID (especially vasculitis and connective tissue diseases) is also considerable in myeloid neoplasms, reaching 20–30% among MDS/CMML patients. AICy are generally related to LPD activity and may worsen the prognosis of the underlying disease, whereas AID may develop before the hematologic neoplasm or during its course, not clearly affecting its prognosis. Likewise, AID-specific treatment may be required independently from the hematologic therapy. It is even more difficult to attribute a pathogenetic, clinical and prognostic value to the several autoimmune serological positivities reported in hematologic neoplasms, as they are not predictive of overt AICy/AID. Notably, a particular attention should be deserved to immune-mediated hemostatic disorders (e.g., AH, TTP, and catastrophic APS), that may jeopardize the onco-hematologic condition. Finally, the new emerging autoimmune complications, mostly AICy in the HSCT and CAR-T cell setting and AID during CPI therapy, highlight that the modulation of the immune system is a complex goal far away to be accomplished.

## Figures and Tables

**Figure 1 cancers-13-01532-f001:**
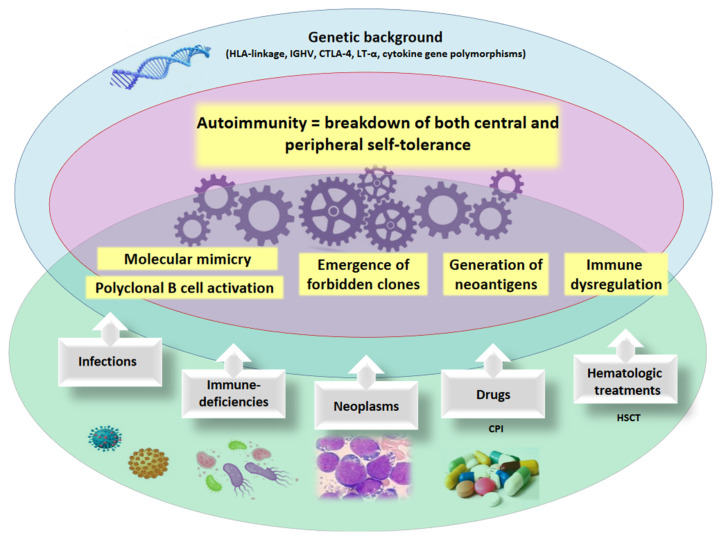
Mechanisms of autoimmunity. The pathogenic mechanisms of autoimmunity involve a genetic susceptibility (i.e., HLA genotype, cytokine polymorphisms, etc.) and the occurrence of acquired, environmental factors (i.e., infectious agents, neoplastic clones, medical/cellular therapies). HLA human leukocyte antigens, IGHV immunoglobulin heavy variable, CTLA-4 cytotoxic T-lymphocyte antigen-4, LT-α lymphotoxin-α, CPI checkpoint inhibitors, HSCT hematopoietic stem cell transplant.

**Table 1 cancers-13-01532-t001:** Autoimmune cytopenias (AICy) complicating lymphoproliferative disorders.

AICy	Lymphoproliferative Disorder	Frequency of AICy	Key Findings	References
AIHA	CLL	7 to 15%	The commonest AICy correlating with advanced disease and high biologic risk (del 11q, del17p, unmutated IGHV)	[1,11]
	NHL	2 to 50%	Frequency is maximal in angioimmunoblastic T-cell lymphoma and in marginal zone lymphoma	[5,12]
	HL	0.2%	Very rare association, but may increase after HSCT or therapy with CPIs.	[4,5,13,14]
	LGLL	Case reports	May be concomitant, precede or follow LGL diagnosis; may be concomitant to ITP and AIN and be multi-refractory (even require splenectomy)	[15]
	CD	Case reports to 6%	May revert after anti-IL6 therapy with tocilizumab	[16]
	ALL	Case reports	Mainly B-ALL in the pediatric setting and post-HSCT	[17]
	MM	Case reports	May rarely complicate MM and also precede the diagnosis.	[18]
ITP	CLL	1 to 5%	Even in association with AIHA (ES). Difficult to distinguish from infiltrative cytopenia; response to steroids and IVIG may confirm the diagnosis	[11]
	NHL	Case reports	Mainly in Waldenström macroglobulinemia and marginal zone NHL; may also follow fludarabine treatment and HSCT.	[12]
	HL	0.2 to 1%	May precede or follow HL diagnosis and be observed even after remission; ITP risk may increase after HSCT or CPIs.	[13,14,19]
	LGLL	1 to 20%	May respond to steroids, IVIG and cytotoxic immunosuppressants used for LGL	[20]
	CD	Case reports	Case reports of ITP and ES during CD progression, may respond to rituximab or be refractory	[21]
	ALL	Case reports	Mostly during chemotherapy. Some require more than 3 lines including splenectomy	[22]
	MM	Case reports	MM may be complicated by ITP and ES; lenalidomide may increase the risk.	[23,24]
AA/PRCA/AIN	CLL	<1%	AIN has been reported in only 3 out of 1750 patients (0.17%); anti-neutrophil autoantibodies may be positive	[11]
	NHL	Case reports	Up to 21 cases of PRCA and rarely AA reported; may develop at onset, during remission, or after chemotherapy and/or HSCT; may be associated with EBV infection and AIHA	[12]
	HL	Case reports	16 patients reported in literature. Case reports of AIN, even years after remission, successfully treated with IVIG	[14,25]
	LGLL	Case reports	Three patients with concomitant amegakaryocytic thrombocytopenia and PRCA and 1 with concomitant AIHA, ITP and AIN	[26]
	ALL	Case reports	Two patients with T-ALL developed PRCA, possibly associated with ALL therapy; 1 patient with T-lymphoblastic lymphoma presented as AA and hypercalcemia	[27]
	MM	Case report	Association of AIN and anti-thyroid autoantibodies	[28]

AIHA autoimmune hemolytic anemia, ITP immune thrombocytopenia, AA aplastic anemia, PRCA pure red cell aplasia, AIN autoimmune neutropenia, CLL chronic lymphocytic leukemia, NHL non-Hodgkin lymphoma, HL Hodgkin lymphoma, LGLL large granular lymphocyte leukemia, CD Castleman disease, ALL acute lymphoblastic leukemia, MM multiple myeloma; HSCT hematopoietic stem cell transplant; CPIs checkpoint inhibitors, ES, Evans syndrome, IVIG intravenous immunoglobulins, EBV Epstein Barr virus.

**Table 2 cancers-13-01532-t002:** Autoimmune diseases (AID) complicating lymphoproliferative disorders.

AID	Lymphoproliferative Disorder	Frequency of AID	Key Findings	References
SLE	CLL	Up to 3%	Various case reports including central nervous system involvement and association with SS	[31]
	NHL	1%	6 SLE out of 612 diffuse large B cell lymphoma cases	[32,33]
	HL	0.02%	1 patient out of 519 HL cases developed SLE	[13]
	CD	0.03% to 1%	9 patients in a systematic review, more than a half had concomitant immune thrombocytopenia; SLE patients had no nervous system involvement	[34]
	LGL	Up to 12%	LGL correlated with > number of SLE exacerbations, cytopenias, and high doses of corticosteroids and immunosuppressors requirement	[35]
	ALL	Case reports	May develop simultaneously or several years after ALL treatment or HSCT	[36]
	MM	Case reports	Either preceding or following MM diagnosis	[37]
RA	CLL	0.4%	Very rare association. Various case reports exist	[31]
	NHL	Case reports to 4%	Oligo- and polyarthritis may occur, mainly associated with T-cell NHL	[32,33]
	HL	Case reports	RA patients with HL seem to have a worse outcome	[13]
	CD	Case reports	Active arthritis is rare in CD patients. Anti-IL6 treatment may be effective	[38]
	LGL	Up to 33%	Felty syndrome (RA, neutropenia, splenomegaly), may benefit from methotrexate (indicated for LGL)	[39]
	ALL	Case reports	Mainly pediatric cases; RA may challenge the differential diagnosis	[38]
	MM	Case reports	Very rare association	[37]
Other AID	CLL	2%	More frequently Hashimoto’s thyroiditis, vasculitis and SS; Case reports of AH, aVWS, and APS	[31]
	NHL	Up to 5%	More frequently SS, but also psoriasis, thyroiditis and Graves’ disease, polymyositis, systemic sclerosis, vasculitis, inflammatory bowel diseases, autoimmune hepatitis, and Addison’s disease. Case reports of AH and aVWS (mainly in MZL and MALT)	[13,32]
	HL	Up to 8.6%	Mainly thyroiditis and Graves’ disease, but also glomerulonephritis, DM type 1, seronegative spondylarthritis, mixed connective tissue disease, systemic sclerosis, and vasculitis. Case reports of catastrophic APS and IgA nephropathy	[13,32]
	CD	Case reports	TAFRO syndrome and concurrent SS; case reports of pemphigus vulgaris and glomerulonephritis	[34]
	LGL	Case reports	SS may complicate up to 25% of T-LGL cases	[39,40]
	ALL	Case reports	Myasthenia gravis; type 1 diabetes mellitus; IgA nephropathy; catastrophic APS	[36]
	MM	Case reports	aVWS and AH, and vasculitis	[41]

SLE systemic lupus erythematosus, RA rheumatoid arthritis, CLL chronic lymphocytic leukemia, NHL non-Hodgkin lymphoma, HL Hodgkin lymphoma, CD Castleman disease, LGL large granular lymphocyte, ALL acute lymphoblastic leukemia, MM multiple myeloma, SS Sjögren’s syndrome, HSCT hematopoietic stem cell transplant, aVWS acquired von Willebrand syndrome, AH acquired haemophilia, APS antiphospholipid syndrome, MZL marginal zone lymphoma, MALT mucosa associated lymphoid tissue, TAFRO thrombocytopenia, anasarca, fever, reticulin fibrosis, and organomegaly.

**Table 3 cancers-13-01532-t003:** Autoimmune cytopenias (AICy) complicating myeloid neoplasms.

AICy	Myeloid Neoplasm	Frequency of AICy	Key Findings	References
AIHA	MDS	0.5–1%	Most cases occur in low-risk MDS, are usually warm AIHA, and may require second- and third-line therapy (CSA, splenectomy, MMF).	[50,51]
	CMML	Case reports	Eculizumab and rituximab have been successfully used.	[52,53]
	CML	Case series	Mostly related to CML therapies in non-transplanted patients (mainly IFN). Described after allogeneic HSCT and related to immune reconstitution, viral infections or CML relapse.	[54]
	Ph-negative MPN	Case reports	Described in primary MF.	[55]
	AML	Case reports	Described in the context of AML secondary to MDS.	[56]
ITP	MDS	Case reports	Generally associated with other autoimmune manifestations (AIHA, SLE, autoimmune nephritis) and sometimes needs second-line treatment (e.g., CSA).	[50]
	CMML	Case series	Generally occurs in CMML-1 along with other autoimmune features (DAT+, ANA+, anti-thyroid Ab+, hypergammaglobulinemia). More frequent in males and elderly. Characterized by higher relapse rate compared with primary ITP. Therapies for idiopathic ITP are effective (except for IvIg). No progression to AML in TPOra-treated patients.	[57,58]
	CML	Case reports	Associated with IFN use.	[59]
	Ph-negative MPN	Case reports	Developed in *JAK2V617F*-mutated ET, apparently not related to hydroxyurea treatment.	[60]
	AML	Case reports	Two cases reported in acute promyelocytic leukemia in complete remission, both steroid-refractory, responded to splenectomy and azathioprine.	[61,62]
AA/PRCA	MDS	Case series	PRCA mostly develops in the context of low-risk MDS with favorable response to IST, whilst efficacy of steroids and rhEPO is poor. May be associated with recurrent cytogenetic abnormalities (e.g., del(5q) and isochromosome 17q).	[63,64,65]
	CMML	Case reports	Generally responsive to steroid +/− CSA. Warnings for possible evolution to AML in CTX- or CSA-treated patients.	[66]
	CML	Case reports	AA associated with imatinib use.	[67]

AIHA autoimmune hemolytic anemia, ITP immune thrombocytopenia, AA aplastic anemia, PRCA pure red cell aplasia, MDS myelodysplastic syndromes, CMML chronic myelomonocytic leukemia, CML chronic myeloid leukemia, Ph-negative MPN Philadelphia-negative myeloproliferative neoplasms, AML acute myeloid leukemia, CSA cyclosporine A, MMF mofetil mycophenolate, IFN interferon, HSCT hematopoietic stem cell transplant, MF myelofibrosis, SLE systemic lupus erythematosus, DAT direct antiglobulin test, ANA anti-nuclear antibodies, Ab antibodies, IvIg intravenous immunoglobulins, TPOra thrombopoietin receptor agonists, ET essential thrombocytemia, IST immunosuppressive treatment, rhEPO recombinant human erythropoietin, CTX cyclophosphamide.

**Table 4 cancers-13-01532-t004:** Autoimmune diseases (AID) complicating myeloid neoplasms.

AID	Myeloid Neoplasm	Frequency of AID	Key Findings	References
Systemic autoimmune disorders	MDS	20–30%	Mainly vasculitis including polyarteritis nodosa, giant cell arteritis, Behçet’s-like vasculitis and other less common types.Discrepancy about the impact of AID on MDS prognosis and survival. Biologics and azacytidine more effective than classic immunosuppressants (steroids and cytotoxic drugs).	[50,68,69,70,71,72,73,74,75]
	CMML	15–25%	AID-CMML patients are younger and mostly CMML-1, show similar AML progression and slightly longer overall survival than non-AID-CMML.Response to steroid is high, but 40–60% of cases need second-line treatment.	[57,68,76]
	Ph-negative MPN	Case reports	Case of polyarteritis nodosa, arthritis, Sjögren syndrome, intestinal autoimmune disorders, dermatomyositis, and multiple sclerosis mainly in MF.	[77,78,79,80]
Other hematologic AIDs	MDS	Case reports	Acquired HA and TTP.	[81,82]
	CMML	Case reports	Acquired HA, TTP, and APS, even catastrophic, either preceding or following CMML diagnosis.	[83,84,85]
	CML	Case reports	TTP and aHUS developed in imatinib- and dasatinib-treated patients.	[86,87]
	Ph-negative MPN	Case reports	Acquired HA, APS, and TTP (in a polycythemia vera patient treated with pegylated interferon).	[88,89]
	AML	Case reports	Acquired HA, acquired factor VII deficiency, fatal catastrophic APS (adult and paediatric, refractory to anticoagulation, plasma exchange and chemotherapy), and TTP.	[90,91,92,93]

AID autoimmune diseases, MDS myelodysplastic syndromes, CMML chronic myelomonocytic leukemia, Ph-negative MPN Philadelphia-negative myeloproliferative neoplasms, AML acute myeloid leukemia, CML chronic myeloid leukemia, MF myelofibrosis, HA hemophilia A, TTP thrombotic thrombocytopenic purpura, APS antiphospholipid syndrome, aHUS atypical hemolytic uremic syndrome.

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
