# Peer review of "Autoimmune Complications in Hematologic Neoplasms"

_cancers, 2021, doi:10.3390/cancers13071532_

Round 1

Reviewer 1 Report

Very well written article, complete and informative.

Author Response

We thank the Reviewer for the positive evaluation.

Reviewer 2 Report

Diligent and comprehensive review manuscript on autoimmune cytopenias and diseases in patients with hematologic malignancies. The review will focus attention of hematologists and oncologists on these frequently overseen complications that may be associated to myeloid or lymphatic malignancies. 

Author Response

(The authors gave the same response as above.)

Reviewer 3 Report

A well written, comprehensive review very useful for the practical hematologist, considering that the situations reported are frequently misinterpreted.

Author Response

(The authors gave the same response as above.)

Reviewer 4 Report

Summary:

  • Review looks at the available literature to define the occurrence of autoimmune cytopenias (AIC) and autoimmune disease (AID) in different hematologic neoplasms
  • AIC occurs in about 10% in CLL with higher frequencies in certain NHLs, where they occur as low as 1% in low risk MDS and CMML. 
  • AID are described up to 30% of myeloid and lymphoid patients, and in approx 10% of patients undergoing HSCT or ICI therapy 
  • I commend the authors on taking on a review of this topic, however significant changes need to be made for this to be acceptable for publication.

Major comments:

  • Authors state that AIC are difficult to diagnosis, can be severe and life threatening, and can have impact on therapies for heme malignancies
  • More evidence is needed for “difficult to diagnose.” They provide some information towards the end of the article regarding autoimmune phenomena that do not reach clinical significance, but do not seem to provide much information regarding patients with clinical autoimmune cytopenias/disease that are difficult to diagnosis. 
  • Good data regarding the impact on therapies/life threatening presentations for myeloid disorders and AID as well as post-HSCT, but this was not present for LPD or AIC. I think this point would need to be changed to reflect what they found
  • Discuss prognostic impact of AIDs and challenges of treatment.

Recommend replacing info on CPI with with the information for other medications treating CLL, CML, MM that have been linked to AIC/AID. CPI and autoimmune effects is its own review and would need to be its own manuscript. Instead, I would focus on post-HSCT induced AIC/AID, as well as specific anti-lymphoid/myeloid malignancy treatment that has been linked to AIC/AID, rather than checkpoint inhibitors.

  • Section 2. Mechanisms of autoimmunity:
  • I would recommend this paragraph to be split in two and discuss the different mechanisms for central and peripheral autoimmunity
  • Discuss the complex interplay between genetic and environmental factors on autoimmunity. I think there is a lot of nice information presented, but it could be condensed and described more succinctly. 
  • Section 3.1.1 AICy in CLL
  • It be important to include any prognostic data regarding their development and outcomes in CLL. From the literature, having autoimmune cytopenias does not portend a inferior prognosis, for example compared to cytopenias due to marrow infiltration from CLL (Hodgson et al Br J Haematol 2011)
  • 1.2 AICy in NHL
  • “It should be noted that cold agglutinin diseases (CAD), sustained by monoclonal cold re-active IgM autoantibodies, is almost invariably associated with a CD5+CD20+ B cell clone, often hardly distinguishable from other NHL, ”
  • a citation needed here to back up this statement
  • 1.3 AICy in HL
  • “Finally, some cases of AIN have been observed in HL, mostly asymptomatic and responding to intravenous immunoglobulins (IVIG) [15]."
  • A new citation is needed here. Source 15 refers to HL patients with aplastic anemia, not autoimmune neutropenia. A search of that reference did not find evidence to support this statement. 
  • 2.1. AID in LGL
  • “In this setting, immunosuppressive therapy may control both diseases, although infectious risk with consequent mortality is very high [30].”
  • This citation I am unable to read the full text of, however the discussion reads: “Some patients with SLE develop LGL proliferations. The activity, clinical severity and hematological involvement seem to be associated with this immunological disorder, but the pathogenic significance and prognosis of these proliferations are still to be elucidated.” I am not sure if this citation really supports the idea that mortality is high w/ immunosuppression for LGL + AID
  • 2.2 AID in CLL
  • “Generally, AH and VWS may respond to CLL therapy including rituximab and venetoclax, whilst the use of ibrutinib is discouraged due to bleeding risk [32,33].”
  • I cannot see the full text for 32, but 33 does not mention this fact, only talks about acquired hemophilia A, and its a general review on hemophilia A not specific for it in CLL, and how it responds to the treatment of CLL. The abstract for citation 32 also does not mention anything about responsiveness of AH and VWS to the treatment of CLL. Need citation to back up this statement. 
  • 2.3 AID in NHL
  • “Finally, case reports are available also for AH and aVWS (mainly in marginal zone lymphoma and mucosal associated lymphoid tissue, MALT) [33]”
  • Citation 33 does not mention anything about acquired von willebrand syndrome and does not mention MALT or MLZ. Need a citation to back up this statement
  • 2.5 AID in other LPD
  • “These cases may confound the initial LPD diagnosis, may be life-threatening (as for cases of catastrophic APS), and may respond to CD and ALL therapies [36-40]."
  • Describing the phenomenon of severity of a catastrophic APS in other LPD, however citations 36-40 are referencing SLE, RA or inflammatory arthritis, not catastrophic APS. Need citation to back up this statement. 
  • Autoimmune complications associated with old and new anti-cancer drugs: “Concerning novel anti-cancer drugs, checkpoint inhibitors (CPIs) reactivate T lym-phocytes to recognize cancer cells by blocking cytotoxic T-lymphocyte antigen-4 (CTLA-4) or programmed death-1 (PD-1) but may induce several immune-related adverse effects."
  • I think the paragraph describing checkpoint inhibitors does not fit into the topic of heme malignancies well because they do not cite references of these drugs being used specifically for HL and leading to higher rates of AIC/AIDs in HL. One of the main citations [101] is a nice meta-analysis regarding the prevalence of autoimmune anemia, neutropenia or thrombocytopenia from checkpoint inhibitors, but these are across solid tumors, a table in the meta-analysis did not list any reports of these side effects in Hodgkin’s patients or other heme malignancies. 
  • This autoimmune phenomenon is obviously of interest, but I do not think it fits with the theme of heme malignancies as this data comes from solid tumors. I think highlighting autoimmunity in post-HSCT and small molecule drugs of CLL, CML, MM, etc. was much more relevant to the overall discussion. 
  • Autoimmune complications associated with old and new anti-cancer drugs: “disorders. An unfavorable outcome is seen in up to 30% of patients and is generally associated with patient comorbidities and NHL stage [105)."
  • Citation 105 refers to neurologic checkpoint inhibitor complications in thoracic malignancies, not NHL or non-Hodgkin lymphoma. 

Minor Corrections

  • Introduction 
  • "Other less investigated autoimmune diseases (AID) include numerous organ-specific disorders targeting endocrine glands, liver, and gut, as well as systemic AID (i.e. systemic lupus erythematosus, SLE, rheumatoid arthritis, RA, and antiphospholipid syndrome, APS)."consider re-wording
  • “In addition, there is evidence of several auto- immune phenomena, i.e. autoantibodies against different autologous proteins, which may anticipate or not overt disease.”
  • Consider re-wording: (may or may not anticipate overt disease)
  • “Such as inhibitors of tumor necrosis factor (TNF)- and other cytokines”
  • Would be more specific, which cytokine inhibitors? Also would change to: “inhibitors of TNF-a, cytokine inhibitors and ICI”
  • Mechanisms of autoimmunity
  • “The former occurs as negative selection by eliminating autoreactive immune effectors during the maturation of the immune system in the bone marrow and the thymus. The second one includes the complex immunologic pathways that may inac-tivate residual autoreactive T cells either by clonal deletion, anergy induction, or conver-sion to regulatory T cells (Tregs). The latter are..."
  • Recommend re-wording, as it sounds confusing. Could say “The former... the latter...and then T-regs instead of the latter again. 
  • "Other key regulators of autoimmunity are antigen presenting cells (APCs), via induction of Treg conversion and interaction with the T cell receptor [6]."
  • Consider re-wording: (APCs), which assist in Treg conversion.
  • How does APC interaction w/ T-cell receptor lead to autoimmunity?
  • Table 1:
  • HL under AIHA and ITP; a “key finding” that AIHA and ITP are rare, but may increase with CPI therapy. Listed next to this key finding is citation [13], which is from 2002 (prior to approval for CPI) and does not mention anything about HSCT and CPI impact on incidence of AICs in HL
  • For Frequency, is this incidence? Please clarify
  •  
  • Figure 1:
  • Would move CPI under drugs, rather than “hematologic treatments” as it is majorly approved for solid tumors but not for hematologic malignancy
  • I think the circles should be moved slightly so it appears as a venn-diagram where the autoimmunity looks as it has occurred due to a complex mix of both genetic factors and environmental factors (infections, immunodeficiency, cancer, drugs, etc)
  • Autoimmune complications in lymphoid neoplasms
  • "As a matter of fact, both AICy and organ/non-organ specific AID, may complicate LPD course from diagnosis to disease pro-gression [11] in up to 50% of cases, although unspecific immune activation has to be dis-tinguished from overt autoimmunity."
  • citation 11 does not appear to be appropriate for thsi statement. Recommend either removing this citation from this area, or adding another citation that reflects this statement. 11 is a paper just discussing AIC and CLL, not all LPD and AIC/AID
  • "In addition, LPD therapy, including HSCT and the use of novel drugs (i.e. checkpoint inhibitors, CPIs) may further trigger autoimmunity, and will be discussed in a dedicated paragraph."
  • Would recommend removal,change to HSCT, CAR-T, idelalisib, imatinib, lenalidomide, etc
  • 1.1 AICy in CLL
  • “In CLL, AIHA is the most frequent form (7-10% of cases), followed by ITP (1-5%), and rarer entities such as pure red cell aplasia (PRCA, <1%) and autoimmune neutropenia (AIN, 0.17%)."
  • Consider re-wording. Would suggest something like “AHIA (7-10%) and ITP (1-5%) occur fairly frequently in CLL, whereare PRCA (<1%) and autoimmune neutropenia (0.17%) are much less common
  • “Importantly, CLL therapy may have an impact on AICy devel-opment: single-agent purine analogs (i.e. fludarabine) may induce AIHA and up to 9% of"
  • I think this could be omitted given that single agent fludarabine is no longer a recommended treatment for CLL, and later in the paper it goes on to reference that FCR shows a decreased rate of AIHA compared to fludarabine alone
  • 1.4 AICy in other LPD
  • “In acute lymphoblastic leukemia (ALL), particularly the T-cell one”
  • Would recommend re-wording
  • Table 1
  • NHL: "Frequency is maximal in angioimmunoblastic T-cell lymphoma and in marginal zone lymphoma", however only cites [12] which is a MZL paper only
  • Need to add additional citations to fit this incidence documented
  • Comma’s that need changed to decimal points
  • Table 2
  • Commas that need changed to decimal points
  • CD row: “9 patients in a systematic review, More than half had concomitant immune thrombocytopenia” 
  • This should belong in table 1
  • Autoimmune complications with old and new anti-cancer drugs
  • “Concerning small molecules such as ibrutinib (that targets Bruton’s tyrosine kinase), idelalisib (that targets phosphoinositide 3-kinase), and venetoclax (a BCL-2 antagonist), no specific trials have studied their role on CLL-AICy."
  • Consider re-wording

Author Response

Please find attached our point-by-point answers.

Round 2

Reviewer 4 Report

Overall, I think the quality of the manuscript has been improved. Minor changes to citations need to be made. Otherwise, I agree the authors responses. See below

1.3 AICy in HL 

“Finally, some cases of AIN have been observed in HL, mostly asymptomatic and responding to intravenous immunoglobulins (IVIG) [15]." 

A new citation is needed here. Source 15 refers to HL patients with aplastic anemia, not autoimmune neutropenia. A search of that reference did not find evidence to support this statement. 

We added the citation required. 

Alliot C, Barrios M, Tabuteau S, Desablens B. Autoimmune cytopenias associated with malignancies and successfully treated with intravenous immune globulins: about two cases. Therapie. 2000 May- Jun;55(3):371-4. PMID: 10967714. 

-----​citation not in references

2.2 AID in CLL 

“Generally, AH and VWS may respond to CLL therapy including rituximab and venetoclax, whilst the use of ibrutinib is discouraged due to bleeding risk [32,33].” 

I cannot see the full text for 32, but 33 does not mention this fact, only talks about acquired hemophilia A, and its a general review on hemophilia A not specific for it in CLL, and how it responds to the treatment of CLL. The abstract for citation 32 also does not mention anything about responsiveness of AH and VWS to the treatment of CLL. Need citation to back up this statement. 

We thank the Referee. We tried to summarize references given the huge amount of literature. In any case, we added the most recent publication about venetoclax effect on aVWS. 

Innocenti I, Morelli F, Autore F, Tomasso A, Corbingi A, Bellesi S, Za T, De Stefano V, Laurenti L. 

Remission of acquired von Willebrand syndrome in a patient with chronic lymphocytic leukemia treated with venetoclax. Leuk Lymphoma. 2019 Dec;60(12):3078-3080. doi: 10.1080/10428194.2019.1612063. Epub 2019 May 20. PMID: 31106622.

-----​citation not in references

2.3 AID in NHL 

“Finally, case reports are available also for AH and aVWS (mainly in marginal zone lymphoma and mucosal associated lymphoid tissue, MALT) [33]” 

Citation 33 does not mention anything about acquired von willebrand syndrome and does not mention MALT or MLZ. Need a citation to back up this statement

The following reference has been added: 

Iwabuchi T, Kimura Y, Suzuki T, Hayashi H, Fujimoto H, Hashimoto Y, Ogawa T, Kusama H, Fukutake K, Ohyashiki K. [Successful treatment with rituximab in a patient with primary thymic MALT lymphoma complicated with acquired von Willebrand syndrome and Sjögren syndrome]. Rinsho Ketsueki. 2011 Apr;52(4):210-5. Japanese. PMID: 21566407.

-----​citation not in references

2.5 AID in other LPD 

“These cases may confound the initial LPD diagnosis, may be life-threatening (as for cases of catastrophic APS), and may respond to CD and ALL therapies [36-40]." 

Describing the phenomenon of severity of a catastrophic APS in other LPD, however citations 36-40 are referencing SLE, RA or inflammatory arthritis, not catastrophic APS. Need citation to back up this statement. 

Agree and added the following citation: 

Miesbach W, Asherson RA, Cervera R, Shoenfeld Y, Gomez Puerta J, Bucciarelli S, Espinoza G, Font J; Members of CAPS Registry Group. The catastrophic antiphospholipid (Asherson's) syndrome and malignancies. Autoimmun Rev. 2006 Dec;6(2):94-7. doi: 10.1016/j.autrev.2006.06.012. Epub 2006 Jul 21. PMID: 17138251. 

-----​citation not in references

We agree that most data about immune related adverse events after CPI come from solid tumor experiences. However, we would like to keep this topic since it is an emerging and often neglected complication also in the hematologic setting (see also previous points). Regarding the mis-citation n.105, we amended the text as follows: 

“Neurologic involvement is rare and included central and peripheral nervous system diseases associated with neural specific autoantibodies, vasculitis, granulomatous and demyelinating disorders, with an unfavorable outcome in up to 30% of patients [105]. 

---in text citation says 109, which should say 105. 

2.1. AID in LGL 

“In this setting, immunosuppressive therapy may control both diseases, although infectious risk with consequent mortality is very high [30].” 

This citation I am unable to read the full text of, however the discussion reads: “Some patients with SLE develop LGL proliferations. The activity, clinical severity and hematological involvement seem to be associated with this immunological disorder, but the pathogenic significance and prognosis of these proliferations are still to be elucidated.” I am not sure if this citation really supports the idea that mortality is high w/ immunosuppression for LGL + AID 

We agree that this does not directly emerge from ref 30, which has now been removed and postponed. In any case, it is well established that the clinical course of Felty syndrome is affected by high infectious risk, which is usually the main cause of death, and that may also be exacerbated by therapeutic immunosuppression.

----​while I do not doubt that felty syndrome can lead to a high mortality from infection, I still believe a citation is needed to back up this comment. 

Author Response

We thank the Reviewer for further revising our manuscript and apologize for the inconvenient. We guess that the manuscript with the revised references was not received by the editorial office. Please find attached the correct version with all the required references and changes highlighted in yellow. Moreover, we added a specific reference for Felty syndrome as suggested.

Best regards
